# An improved nucleic acid sequence-based amplification method mediated by T4 gene 32 protein

Yi Heng Nai⬤*, Egan H. Doeven, Rosanne M. Guijt⬤*

Centre for Regional and Rural Futures, Deakin University, Geelong, Victoria, Australia

* ryan.nai@gmail.com (YHN); rosanne.guijt@deakin.edu.au (RMG)

**Data Availability Statement:** All relevant data are within the manuscript and its Supporting information files.

**Funding:** The author(s) received no specific funding for this work.

## Abstract

The uptake of Nucleic Acid Sequence-Based Amplification (NASBA) for point of care testing may be hindered by a complexity in the workflow due the requirement of a thermal denaturation step to initiate the cyclic isothermal amplification before the addition of the amplification enzymes. Despite reports of successful enhancement of other DNA and RNA amplification methods using DNA and RNA binding proteins, this has not been reported for NASBA. Here, three single-stranded binding proteins, RecA, Extreme Thermostable Single-stranded binding protein (ET SSB) and T4 gene gp32 protein (gp32), were incorporated in NASBA protocol and used for single pot, one-step NASBA at 41 ˚C. Indeed, all SSBs showed significantly improved amplifications compared with the 2-step process, but only gp32 showed no non-specific aberrant amplification, and slightly improved the time-to-positivity in comparison with the conventional NASBA. For synthetic HIV-1 RNA, gp32 was found to improve the time-to-positivity (ttp) by average of 13.6% of one-step NASBA and 6.7% of conventional NASBA for the detection of HIV-1 RNA, showing its potential for simplifying the workflow as desirable for point of care applications of NASBA.

## Introduction

Nucleic acid amplification testing (NAAT) is used for screening and identification of a pathogen for the diagnosis of pathogen-based infections in the clinical, veterinary and agricultural sectors. Polymerase Chain Reaction (PCR) is the gold standard methodology widely employed in centralized diagnostic laboratories. Its operational requirement for thermocycling, however, imposes engineering challenges for the deployment of PCR in point of care testing [1]. Commercial products on the market demonstrate these challenges have been successfully overcome, but elegant alternatives eliminating the need for thermocycling by adopting isothermal amplification methods are rapidly gaining popularity. Different isothermal amplification methods have been developed and adopted for clinical diagnostics, demonstrating efficient and fast amplification at a constant temperature [2].

In PCR, the elevated temperature step denatures double-stranded DNA (dsDNA) as required for primer annealing and extension. Isothermal amplification technologies employ

**Competing interests:** The authors have declared that no competing interests exist.

different strategies to achieve this, including enzymatic activities or primer design to access dsDNA. Generally, primer binding sites are the initiation sites for DNA polymerases with strand displacement activity (e.g. large fragment of *Bsu*, *Bst*, and E. coli DNA Polymerase I, or phi29 DNA polymerase) leading to dsDNA separation and extension yielding single-stranded DNA to initiate the isothermal amplification [1, 2]. Isothermal methods vary in the way initiation is facilitated, with strategies categorized as (i) strand invasion facilitated by recombinases and single-stranded binding protein (SSB) in Helicase Dependent Amplification (HDA) [3], Strand Invasion Based Amplification (SIBA) [4] and Recombinase Polymerase Amplification [5], (ii) thermodynamic invasion and primer annealing in Nucleic Acid Sequence Based Amplification (NASBA) [6], transcription-mediated amplification (TMA), Loop mediated isothermal amplification (LAMP) [7, 8], (iii) the use of nicking enzymes in Strand Displacement Amplification (SDA) [9] and Nicking Enzyme Amplification Reaction (NEAR) [10].

NASBA is primarily employed to directly amplify RNA targets and—using a molecular beacon for detection—it was demonstrated to be less prone to false positives in the presence of genomic DNA [11, 12] and faster than alternatives relying on reverse transcription of the RNA into DNA without the need for DNase I treatment. A schematic representation of NASBA amplification is provided in Fig 1. In conventional NASBA process, a pre-denaturation step of the RNA at 65 ˚C is required, followed by annealing of the primers to the corresponding RNA template and addition of NASBA enzyme cocktail: Avian myeloblastosis virus reverse transcriptase (AMV RT), RNase H, and T7 RNA polymerase to kickstart the amplification at 41

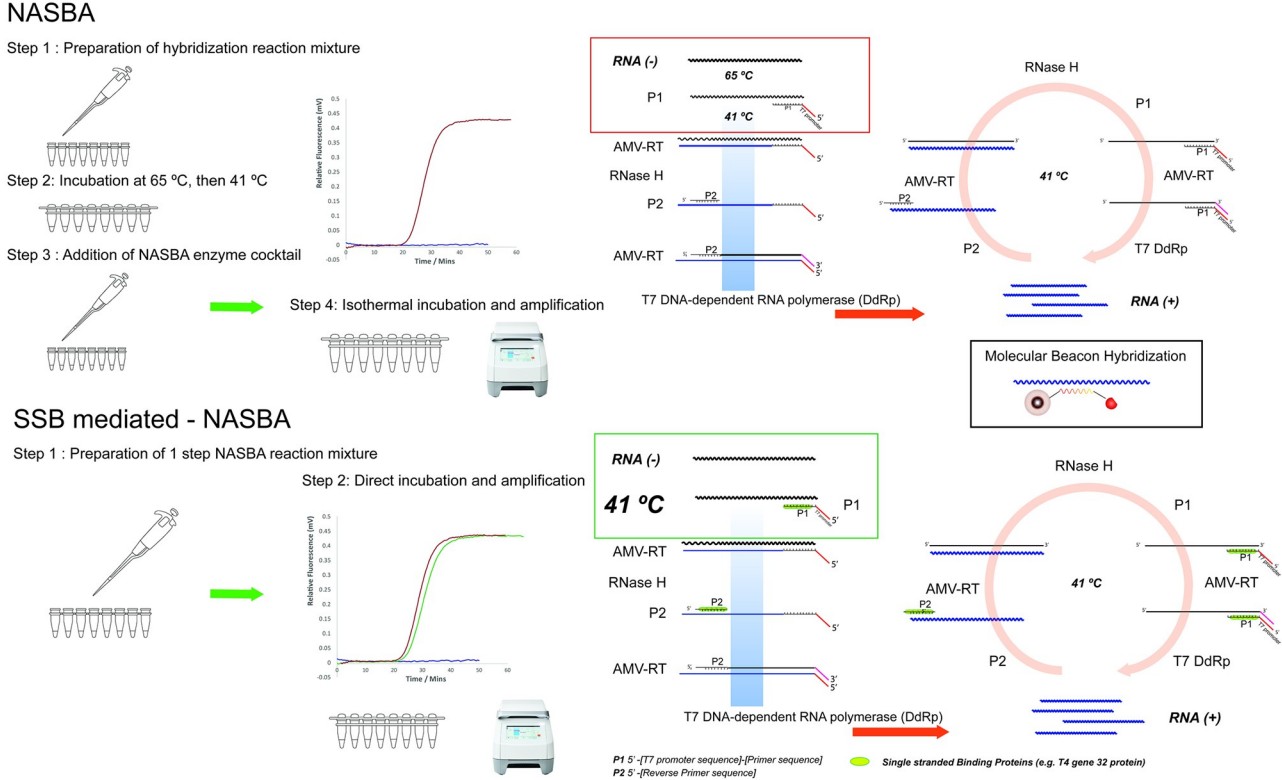

**Fig 1. A schematic representation of conventional NASBA and SSB mediated NASBA.** The conventional NASBA protocol requires a hybridization step before the addition of thermolabile NASBA enzymes to initiate the amplification and signal acquisition measuring the fluorescence intensity of molecular beacon. In single stranded binding protein mediated NASBA, the thermal denaturation-hybrization step is circumvented, allowing truly isothermal, single pot NASBA. In the graphic representation of the amplification cycle, the solid lines represent DNA, the wavey lines RNA.

˚C. Because T7 RNA polymerase is thermolabile, NASBA is executed as a 2-step process, adding the enzymes after the thermal primer annealing step [13]. Recognizing NASBA and TMA are less frequently used than PCR, the assays are performed many thousands to millions of times a day in platforms including bioMérieux Nuclisens® and Hologic Panther®. Compared to other isothermal amplification methods, such as LAMP or Recombinase Polymerase Amplification (RPA), the 2-step process could be considered inconvenient for high throughput or point of care operation as each additional processing step may present an operational and engineering challenge [14]. Procedural simplicity is critical reduce cost, prevent malfunction and achieve robust, reliable operation [1]. A recent review on enhancement strategies for isothermal amplifications concluded that effective additives used to eliminate the thermal denaturation step include helicase, recombinases, endonucleases, ionic liquids, betaine, proline and trehalose [15], and SSBs used for decreasing non-specific interactions and to prevent nucleic acid degradation. Here, the potential of SSBs to facilitate primer binding and eliminate the thermal annealing step in NASBA is reported as enhancement strategy, facilitating NASBA as a single step method for enhanced field deployability.

## Results and discussion

A truly isothermal, single-step NASBA method is anticipated to enhance NASBA's potential for adoption in portable diagnostic devices, as it eliminates the need for complex temperature and fluidic control, hence simplifying instrument and operational design. In addition to their role decreasing non-specific interactions [15], SSBs have been employed to facilitate strand invasion and initiate amplification in various isothermal amplification methods [16, 17]. This evidence suggests SSBs may also be employed to avoid the thermal pre-denaturation step in NASBA. Here, 3 commercially available SSBs, namely *E. coli* RecA, Extreme Thermostable Single-stranded binding protein (ET SSB), and T4 gene 32 protein (gp32) were examined to assess their ability to substitute the thermal pre-denaturation step and simplify NASBA method (Fig 1). A myriad of techniques has been developed for the detection and quantification of nucleic acid amplification products, ranging from simple colorimetric reactions for visual read-out to more complex electrochemical and optical sensors; a comprehensive review focusing on point of care analysis can be found elsewhere [18]. Here, we used the fluorescence signal from a molecular beacon for fluorescence detection.

To verify the hypothesis that SSBs could facilitate the conventional 2-step NASBA to take place as a single pot reaction, the SSBs were incorporated into NASBA amplification mixtures (LRB-5, Life Sciences Advanced Technologies LLC, FL, USA), and amplification rate and yield were evaluated. Briefly, the reaction mixture contained primers, target RNA, NASBA enzymes (LEM-5) and molecular beacon for detection, all in NASBA reaction buffer. SSBs were added to the amplification mix at amount of 2 μg of RecA, 80 ng ET SSB, or gp32 (100–320 ng/μL; 1.5–9.6 μM) per reaction respectively. Unless specified, single-step NASBA reactions were performed at 41 ˚C for the one-step amplification of the synthetic HIV-1 gag RNA, one of NASBAs commercial targets with its HIV-1 gag gene primer set (S1 Table in S1 File) without the heat denaturation step in a qPCR machine (Biorad CFX Connect) with fluorescence signal acquisition at 1 min interval for 60 minutes. The resulting graphs for ET SSB and RecA and gp32 are presented in Fig 2a–2c. Please note the recorded amplification times do not include the extra time required for the addition of the amplification enzymes after thermal annealing using the 2-step protocol. Attenuated amplification was observed for all one step NASBA reactions.

Data in Fig 2. confirm SSBs can be used to improve the time-to-positivity (ttp) of one-step NASBA at 41 ˚C, with varying effect on the amplification kinetics and specificity. The addition

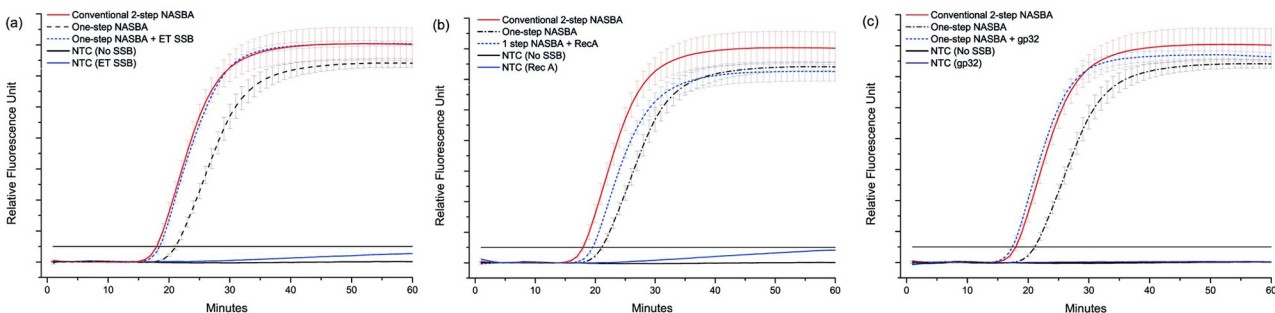

**Fig 2. The effect of SSBs on NASBA of the synthetic HIV-1 RNA gene using (a) 80 μg ET SSB, (b) 2 μg RecA and (c) 1 μg gp32 of a sample containing 2 x 10$^5$ copies of synthetic HIV-1 RNA (~C$_t$ 23) using the HIV-1 NASBA primer-probe set (S1 File).** All graphs show conventional 2-step NASBA (red), 1-step NASBA (black dash/dot), 1-step NASBA with SSB (blue dashed), and NTC in absence (solid black) and presence (solid blue) of SSB (n = 3).

of ET SSB resulted in an increase in amplification rate matching the conventional NASBA at high RNA target input (2 x 10$^5$ copies, ~ C$_t$ value of 23). Interestingly, the fluorescence intensity for the no template control (NTC) was also increased, indicating either a loss in specificity or aberrant collateral interaction with molecular beacon and hence increased risk of false positives past 60 min. In the case of RecA a slight improvement in the amplification rate compared to one-step NASBA was found, but the amplification was attenuated compared with conventional NASBA. Similarly to ET SSB, a slight increase in the NTC signal was observed in the presence of RecA. This only occurred after extended amplification times and is unlikely to cause false positives unless incubation times > 60 min are used (data not shown), but it does indicate a decrease in specificity of the amplification. RecA has been demonstrated to enhance fidelity in multiplex PCR [19] and LAMP [20], RecA was not investigated further.

The addition of gp32 improved one-step NASBA (1 μg), with the ttp of 17.3 ± 0.36 min for the one-step reaction comparable to the 18 ±0.57 min ttp of conventional 2-step process. Importantly, no increase in NTC signal was observed, suggesting no adverse impact on specificity. To our best knowledge, gp32 has not been reported for NASBA but has been assigned multifaceted roles in various DNA and RNA amplifications. When incorporated in PCR, gp32 increased amplicon length, PCR yield [21] and DNA sequencing read length [22] and alleviated PCR inhibition [23]. In T4 bacteriophage DNA replication, gp32 binding of ssDNA was not sequence-specific [24–26] and it facilitates replication, recombination and repair by delivering unfolded DNA to the respective enzymes [24]. In RPA and SIBA [4, 5] gp32 was demonstrated to aid recombinases UvsX and UvsY facilitating primer annealing by invading dsDNA targets, enabling exponential DNA amplification. Besides, gp32 enhances polymerase and RT activities by preventing the formation of secondary structures of template ssDNA and RNA, respectively, allowing for doubling of the yield of in vitro transcripts by T7 RNA polymerase and a significant increase for reverse transcriptase [27]. More recently, gp32 has shown to assist strand invasion of primers at moderate temperatures (30–45 ˚C) and to facilitate the formation of an initiation site for DNA and RNA amplification [4, 16, 25]. Based on the different roles gp32 has played in amplification, we propose its benefits in NASBA may be multifaceted. Its ability to stabilize displaced ssDNA may improve primer annealing in NASBA by stabilizing the single-stranded cDNA formed by AMV RT.

Recognizing further investigation will be required to elucidate the mechanistic role(s) gp32 plays in facilitating the initial cyclic amplification step of dsDNA by T7 polymerase in NASBA, the presented results do demonstrate gp32 can facilitate one-step NASBA detection of HIV-

1with slightly improved ttp in comparison with the 2-step protocol, eliminating the need for thermal pre-denaturation. The effect of gp32 concentration in the amplification reaction on ttp was investigated, with results summarized in Fig 3a. The optimum performance of gp32 was observed at concentrations between 3–4.5 μM (~100–150 ng/μL), with ttp faster than both conventional and no SSB one-step NASBA procedure, yet the ttp gradually increased at higher concentrations suggesting a concentration dependent effect. Using 3 μM of gp32 as the optimum for HIV-1 gag gene primer set, we observed similar improvement for various RNA inputs (Fig 3b). With mechanistic detail of gp32s effect in one-pot NASBA is still unclear, the attenuating effect of higher levels of gp32 should be investigated further. To the authors best knowledge, there are no literature reports of single step NASBA protocols, but anecdotal reports confirmed in our laboratory suggest the reliance on the thermal annealing step varies for primer/targets. The effect of gp32 was studied using the Acrometrix HIV-1 control, and a near-negligible improvement in ttp (from 29 min to 28.5 min) was found using this full length HIV-1 RNA (S1 File), a result that may have been caused by the low target concentration (100 copies/reaction). Improvements in ttp using gp32 were found for Avian influenza A H5N1 RNA, decreasing ttp from 38 to 31 min in presence of gp32, compared with a ttp of 28 min for the two-step process (unpublished results), confirming the performance enhancement with another primer set.

In conclusion, SSBs can be used to decrease the reliance on a thermal annealing step in NASBA, enhancing the procedural simplicity by facilitating one-pot NASBA. ET SSB, RecA and gp32 were evaluated, and gp32 was selected for further study as it provided faster amplification and no non-specific amplification. One-pot amplification of the HIV-1 gag gene mediated by 3 μM gp32 enabled a moderate ttp improvement in average of 6 min in one-step NASBA and 2.4 min in conventional NASBA (not accounting the manual addition of amplification enzymes after primer annealing). The effects were confirmed across VLP RNA input levels from $C_t$ 20 to $C_t$ 34, with improvement in amplification across the studied range. While mechanistic studies explaining the role of gp32 in one step NASBA, as well as to examine the effect of the target and primer are required, we believe the results obtained using gp32 can reposition NASBA in clinical diagnostics, particularly for tests executed outside the laboratory setting, as COVID-19 pandemic has demonstrated the need for fast, decentralized NAAT.

## Material and methods

### NASBA and purification of target RNA amplicon

The two target RNA samples used in the NASBA study were purified from synthetic viral-like particles (VLP) RNA. We packaged artificial sequence into MS2 capsid, which created the VLPs to serve as a model for carrying target sequence of interest in this study (HIV-1 gag gene Genbank AntiSense, strain HXB2, Accession #k03455, nt 1359–1499). The VLP was prepared as previously described [28]. RNA purification was performed using spin column based Macherey-Nagel NucleoSpin RNA Virus Mini kit (catalog Number 740956.50, Scientifix Pty Ltd, Victoria, Australia) as per the manufacturer's protocol. The amplification process was performed using commercial NASBA reagent from Life Sciences Advanced Technologies, Inc. (St. Petersburg, FL, USA). Briefly, 15 μL reaction buffer mixture (LRB) consists of 40 mM Tris HCl (pH 8.5), 70 mM KCl, 12 mM MgCl₂, 15% dimethyl sulfoxide, 5 mM dithiothreitol (DTT), 1 mM dNTP mixture, 2 mM ATP, CTP and UTP mixture, 1.5 mM GTP, 0.5 mM ITP, 0.1 μM molecular beacon probe, 0.2 μM of P1 and P2 primers (Integrated DNA Technologies, IL, USA) and purified RNA.

In conventional NASBA, the reaction mixture was preincubated at 65 ˚C for 2 min then lowered to 41 ˚C, amplification reaction was initiated by addition of 5 μL of enzyme cocktail

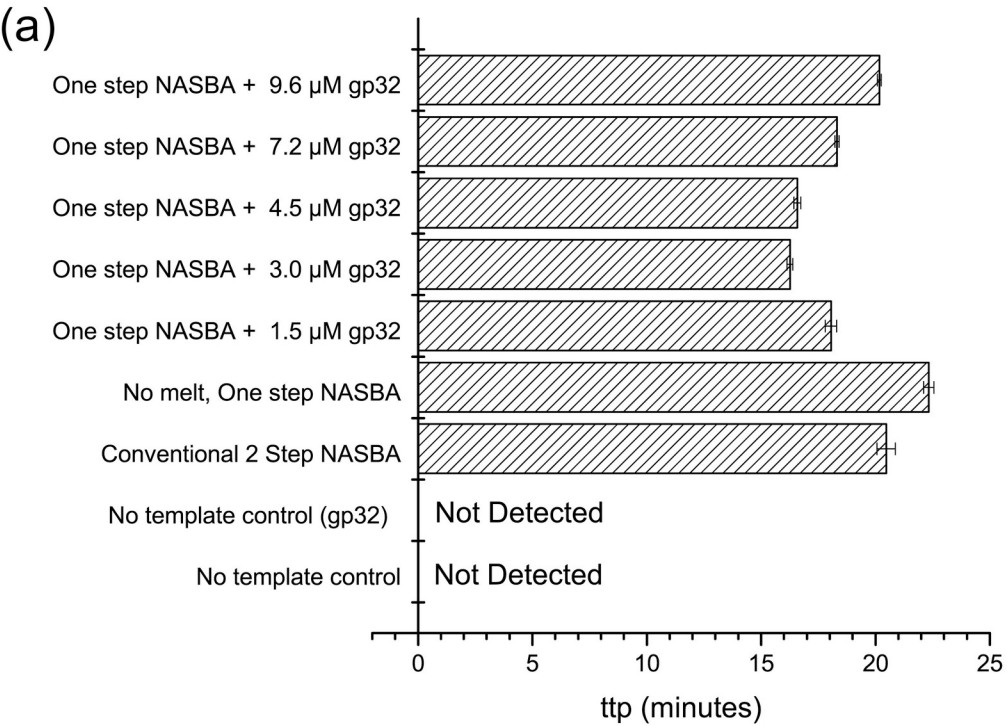

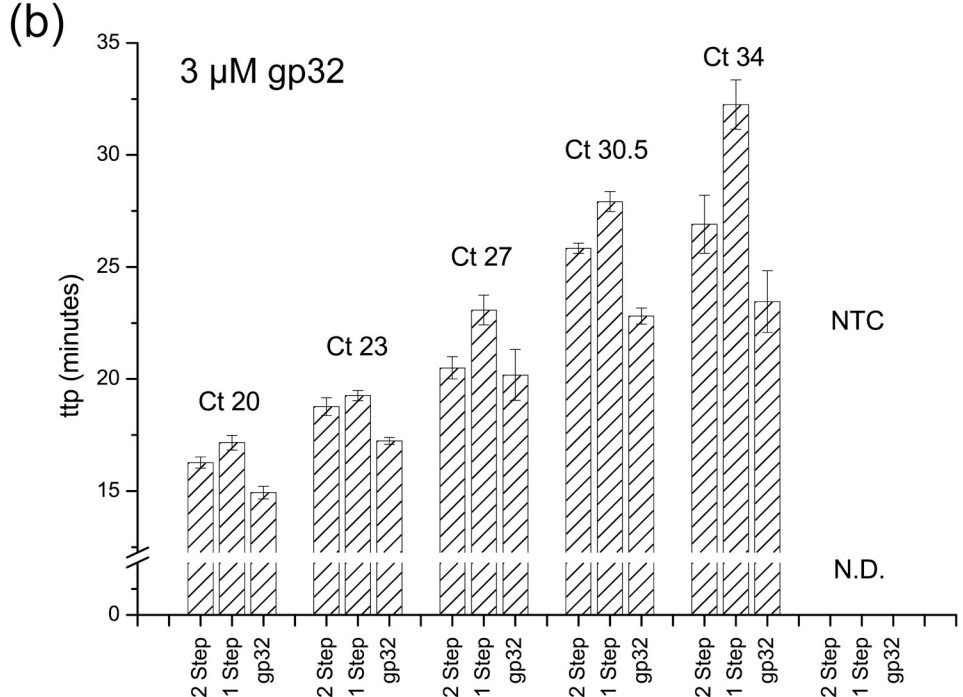

**Fig 3.** (a) Effect of gp32 protein on the ttp for HIV-1 using NASBA and HIV-1 primer set (S1 File) (n = 3). (a) Optimization of gp32 using 0, 3, 4.5, 7.2 and 9.6 μM (0, 100, 150, 240, 320 ng/μL) gp32. Sample contained ~1 x $10^5$ copies of synthetic HIV-1 RNA; conventional 2-step NASBA included as reference, NTCs as 2-step (no gp32) and 1-step (with 4.5 μM gp32) (b) Comparison of NASBA carried out as conventional 2-step, one step in absence of SSB, and 1-step using 3 μM gp32 for HIV-1 RNA samples ranging from $C_t$ 20 (~1.5 x $10^6$ copies) to $C_t$ 34 (~1.5 x $10^2$ copies).

(LEM) containing three enzymes, namely 6.4 U AMV Reverse Transcriptase (AMV-RT), 32 U T7 RNA polymerase, and 0.1 U ribonuclease H. NASBA reaction tubes were incubated at 41˚C for 60 min with signal acquisition at 1 min interval. NASBA primers [29] and RNA target sequence are detailed in S1 Table in S1 File.

In SSB mediated NASBA, SSBs were added to the amplification mix in combination with the LEM cocktail at 2 μg of RecA (M0249, New England Biolabs (NEB)), 80 ng ET SSB (M2401S, NEB) or gp32 (M0300, NEB) at a concentration range between 50–320 ng/μL per reaction respectively. The amplification and signal acquisition were initiated and carried out at 41 ˚C for 90 min without the 65 ˚C melting step.

## Nucleic acid extraction and qPCR quantitation

The RNA purification of VLP RNA and Thermo Scientific™ AcroMetrix™ HIV-1 Control were performed according to kit protocol yielding 50 μL of purified nucleic acid extract. The purified RNA was serially diluted linearly by 10-fold in RNase-Free Tris EDTA (pH 7.5) buffer, and the quantitation of the series was established using qRT-PCR using hydrolysis probe method (Bioline SensiFAST™ Probe No-ROX One-Step Kit, Bioline AUS, Sydney) with HIV-1 gag primer-probe set (S1 Table in S1 File). Subsequently, the calibrated nucleic acid extracts (S1 File) were transferred to an 8-well strip of 0.2 ml polypropylene tubes at stored under -80 ˚C until required.

## Supporting information

**S1 File. S1 Table and information of VLP RNA quantification.**
(DOCX)

## Author Contributions

**Conceptualization:** Yi Heng Nai.

**Investigation:** Yi Heng Nai.

**Methodology:** Yi Heng Nai, Egan H. Doeven.

**Supervision:** Rosanne M. Guijt.

**Writing – original draft:** Yi Heng Nai.

**Writing – review & editing:** Egan H. Doeven, Rosanne M. Guijt.

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
