## [Decision Letter · Decision Letter 0]

4 May 2021

PONE-D-21-08924

An Improved Nucleic Acid Based Sequence Amplification Method Mediated by T4 Gene 32 Protein

PLOS ONE

Dear Dr. Nai,

Thank you for submitting your manuscript to PLOS ONE. After careful consideration, we feel that it has merit but does not fully meet PLOS ONE’s publication criteria as it currently stands. Therefore, we invite you to submit a revised version of the manuscript that addresses the points raised during the review process.

We look forward to receiving your revised manuscript.

Kind regards,

Alberto Amato

Academic Editor

PLOS ONE

Additional Editor Comments:

The reviewers have identified a number of weak points which need your attention.

Journal Requirements:

Reviewers' comments:

Reviewer's Responses to Questions

**Comments to the Author**

1. Is the manuscript technically sound, and do the data support the conclusions?

Reviewer #1: Partly

Reviewer #2: Partly

2. Has the statistical analysis been performed appropriately and rigorously? 

Reviewer #1: No

Reviewer #2: No

3. Have the authors made all data underlying the findings in their manuscript fully available?

Reviewer #1: Yes

Reviewer #2: Yes

4. Is the manuscript presented in an intelligible fashion and written in standard English?

Reviewer #1: Yes

Reviewer #2: No

5. Review Comments to the Author

Reviewer #1: The manuscript from Nai, Doeven and Guijt describes an improvement to NASBA reactions by inclusion of single-stranded binding protein, particularly T4 gp32. This would be a good benefit to improving usability and application of NASBA, but I have a few points that I think should be addressed before acceptance for publication.

1) As the authors describe, an initial denaturation step at 60-65C is typically done before adding NASBA enzymes and running the detection reaction. But the reason for this step is pretty fundamental, typically NASBA reactions do not work at all without it. Here the data show that the "one-step" protocol is actually just fine, a few minutes slower than the standard two-step but otherwise amplifying and detecting target well. This is an unusual result and I wonder if the simplicity of the used template or something else about the particular reaction being done here gives rise to an abnormal finding. If the key bottleneck for wider use of NASBA is this heat step, it seems from the data here that that step is already unnecessary. It would be good to investigate other primers/targets to demonstrate that the gp32 effect is not assay-specific, but it could also add to this point which is very unexpected to me.

2) The authors describe better usability of NASBA for point-of-care settings, but the reaction requires a molecular beacon and real-time fluorescence as well as extracted RNA. Could the detection be done more simply? Could the VLPs be used directly without extracting RNA? I also don't see the benefit of encapsulating a synthetic RNA then extracting it again, if a mock sample is desired here then a real mock sample should be used, spiking the VLPs into a swab or blood or whatever.

3) Some statistical analysis would help to show that the benefits/differences between the conditions are significant. The authors at one point compare 17.3 and 18 minutes, for example, and the data in Figure 3 would benefit from some significance determination. In Figure 3 the error bars in (a) seem extremely small compared to (b), the number of replicates is not listed and I find the difference odd.

4) As the authors note there are other isothermal methods being more widely used, RPA, LAMP, etc. What's the benefit of NASBA over those? Is it just that NASBA is an alternative, or does it offer something else? Sensitivity shown here is pretty moderate and I'd think any of the methods could pick up 150 copies in 30 minutes, and some without requiring fluorescence detection. Some commentary on why NASBA would be useful, even if it's just "as COVID has demonstrated we need as many methods as possible".

4) Minor points:

It would be good to show the molecular beacon binding the target in Figure 1. Also it's not very clear which strands are DNA and which are RNA as the images and labels are very small.

NASBA and TMA are of course not obscure, rarely-used methods but diagnostic workhorse methods performed by the many thousands or millions every day in platforms like the Hologic Panther. Would be good to note this, and that the 2-step temperature requirement isn't a problem for widespread use just for simple applications.

In the introduction the authors state that NASBA is "less prone to false positives and faster than alternatives relying on reverse transcription of the RNA into DNA." NASBA is only less prone to false positives with molecular beacon detection, certainly not if using an intercalating dye or other readout, and it of course requires reverse transcription of the RNA into DNA otherwise it doesn't work as the authors' own cartoon shows. This should be restated.

Reviewer #2: Title: An Improved Nucleic Acid Based Sequence Amplification Method Mediated by T4 Gene 32 Protein

Manuscript ID: PONE-D-21-08924

Authors: Yi Heng Nai*, Egan H. Doeven, and Rosanne M. Guijt

Submitted to PLOS ONE

This manuscript describes an improved nucleic acid sequence-based amplification (NASBA) using single-stranded binding protein (SSB), T4 gene protein 32 (T4gp32). In this strategy, three SSBs were employed to construct a modified NASBA capable of eliminating the initial denaturation step and thus amplifying target RNA molecule in a single pot at 41 ℃. The authors applied this strategy to amplify synthetic HIV-1 RNA molecules. The initial denaturation step in the traditional NASBA is needed to disrupt the complicated secondary structure of long genomic RNAs and the authors should validate the benefits of this improved NASBA technology by amplifying long genomic RNAs but not short synthetic RNAs which might rarely require the initial denaturation. Furthermore, this manuscript is not well-structured and contains a lot of errors. Therefore, I would not recommend acceptance of this manuscript. Some of my other comments are as follows.

Comment 1. Nucleic Acid Based Sequence Amplification’ should be corrected to ‘Nucleic Acid Sequence-Based Amplification (NASBA)’ throughout the overall manuscript.

Comment 2. Many abbreviations such as ttp and ET SSB are not defined.

Comment 3. The authors are recommended to provide standard deviation for Ct data in Figure 2.

Comment 4. The authors need to present the experimental data obtained from RT-qPCR conducted to confirm the concentrations of target RNA.

Comment 5. The authors need to provide relevant references

1) Related to false positives of the NASBA reaction. (Page3, Line 59)

2) Related to the thermolability of the T7 RNA polymerase. (Page 4, Line 65)

3) Related to the T4gp32 effects including the increase of the DNA sequencing read length and alleviation of the PCR inhibition. (Page 5, Line 106)

6. PLOS authors have the option to publish the peer review history of their article (what does this mean?). If published, this will include your full peer review and any attached files.

Reviewer #1: No

Reviewer #2: No

---

## [Author Response · Author response to Decision Letter 0]

4 Nov 2021

Dear Editor,

Please find enclosed submission with our revised manuscript as per instructed in the decision letter.

1) Addressing reviewer comments can be found in the document - 'Response to Reviewers'.

2) A marked-up copy of manuscript that highlights changes made to the original version.

An unmarked version of your revised paper without tracked changes.

We look forward to receiving a favourable outcome on this resubmission.

Regards

Yi Heng Nai

---

## [Decision Letter · Decision Letter 1]

20 Dec 2021

PONE-D-21-08924R1An Improved Nucleic Acid Sequence-Based Amplification Method Mediated by T4 Gene 32 ProteinPLOS ONE

Dear Dr. Guijt,

Thank you for submitting your manuscript to PLOS ONE. After careful consideration, we feel that it has merit but does not fully meet PLOS ONE’s publication criteria as it currently stands. Therefore, we invite you to submit a revised version of the manuscript that addresses the points raised during the review process.

We look forward to receiving your revised manuscript.

Kind regards,

Alberto Amato

Academic Editor

PLOS ONE

Journal Requirements:

Additional Editor Comments (if provided):

Please take into consideration all the reviewrs' comments

Reviewers' comments:

Reviewer's Responses to Questions

**Comments to the Author**

1. If the authors have adequately addressed your comments raised in a previous round of review and you feel that this manuscript is now acceptable for publication, you may indicate that here to bypass the “Comments to the Author” section, enter your conflict of interest statement in the “Confidential to Editor” section, and submit your "Accept" recommendation.

Reviewer #1: (No Response)

Reviewer #3: All comments have been addressed

2. Is the manuscript technically sound, and do the data support the conclusions?

Reviewer #1: Partly

Reviewer #3: Yes

3. Has the statistical analysis been performed appropriately and rigorously? 

Reviewer #1: No

Reviewer #3: Yes

4. Have the authors made all data underlying the findings in their manuscript fully available?

Reviewer #1: Yes

Reviewer #3: Yes

5. Is the manuscript presented in an intelligible fashion and written in standard English?

Reviewer #1: Yes

Reviewer #3: Yes

6. Review Comments to the Author

Reviewer #1: I thank the authors for addressing the comments from the reviewers, the resulting updated version here is better but I still have a fundamental point that now if anything needs more attention. Also my intent from the first round was to get some of the points raised added into the manuscript, the authors seemed to disagree with me but I maintain that the things raised in the first review are still relevant, if no experiment is required just some discussion or text I fail to see how that is "out of scope of the current communication.

But more importantly, both I and reviewer #2 pointed out that the thermal denaturation/annealing step is generally considered required for NASBA/TMA to work well. The authors are claiming it is not when gp32 is present. That would be great, so in the assay shown matching the speed of the typical 2-step protocol just by including gp32 is a nice result...but it is very plainly a result of using this specific assay, as the 1-step, non-gp32 version works surprisingly well. The authors own data in the rebuttal shows another amplicon where the 1-step gives very poor signal and the gp32 addition does not rescue it to the level of the 2-step, let alone faster as they (unconvincingly from the limited data points) claim in the manuscript. I feel this proves my point, not theirs, and the gp32/1-step effect may indeed by an artifact of the assay used in the main body of the text. 2 assays are shown here, and the authors' claims are only true for 1 of them. Either more primer/sets assays should be shown where the gp32/1-step is indeed as good as the 2-step, or the authors should modify the claims in the manuscript to be more honest and in line with the data. That in circumstances where avoiding heat denaturation is paramount to NASBA utilization, then gp32 may help, but there may be sacrifices in performance depending on the assay. That is obviously not a barrier to high-throughput NASBA/TMA as the authors now even state, the Hologic and Biomerieux platforms do seem to work. Perhaps a simple point-of-care or at-home test NASBA would indeed be more practical if the 2-step protocol can be avoided, and the manuscript would be stronger if the authors present that more honestly.

Reviewer #3: Authors of the manuscript entitled “An Improved Nucleic Acid Sequence-Based

Amplification Method Mediated by T4 Gene 32 Protein” have thoroughly revised the said

manuscript in the light of comments/suggestions raised by the reviewers. Responses to

reviewers’ comments have satisfactorily been addressed by the authors and have also been

incorporated at the appropriate places within the revised manuscript. The revised

manuscript may now be accepted for publication in PLOS ONE. However, there is a couple of

very minor corrections that should be done at the Journal level before publication.

1. Introduction section on page no. 4, line no. 73- abbreviation ‘RPA’ has been used for

the first time in the manuscript, so its full form ‘Recombinase Polymerase

Amplification’ should also be written at that place.

2. Results and Discussion section, page 5, line 97- ‘times to not include’ should be

‘times do not include’.

7. PLOS authors have the option to publish the peer review history of their article (what does this mean?). If published, this will include your full peer review and any attached files.

Reviewer #1: No

Reviewer #3: No

---

## [Author Response · Author response to Decision Letter 1]

15 Feb 2022

Reviewer #1: I thank the authors for addressing the comments from the reviewers, the resulting updated version here is better but I still have a fundamental point that now if anything needs more attention. Also my intent from the first round was to get some of the points raised added into the manuscript, the authors seemed to disagree with me but I maintain that the things raised in the first review are still relevant, if no experiment is required just some discussion or text I fail to see how that is "out of scope of the current communication.

Response

We would like to apologise not having addressed detection in detail.

Action taken: 

A sentence and reference on detection have now been included from line 92:

“A myriad of techniques has been developed for the detection and quantification of nucleic acid amplification products, ranges from simple colorimetric reactions for visual read-out to more complex electrochemical and optical sensors; a comprehensive review focusing on point of care analysis can be found elsewhere [18]. Here, we used the fluorescence signal from a molecular beacon for fluorescence detection.”

But more importantly, both I and reviewer #2 pointed out that the thermal denaturation/annealing step is generally considered required for NASBA/TMA to work well. The authors are claiming it is not when gp32 is present. That would be great, so in the assay shown matching the speed of the typical 2-step protocol just by including gp32 is a nice result...but it is very plainly a result of using this specific assay, as the 1-step, non-gp32 version works surprisingly well. 

Response:

We agree that for HIV-1, the one step protocol worked surprisingly well, a phenomenon occasionally observed (but not reported) for NASBA amplifications. Unfortunately, we have been unable to identify “typical” attenuation for one step NASBA reactions for benchmarking purposes, as articles typically refer to the protocols developed in the 1990’s without presenting the attenuation experienced. We hypothesize it depends on the primers used as well as the formulation of the amplification buffer. A discussion of the observed amplification in single pot NASBA has now been included (see below under action taken). 

The presented research shows that T4GP32 outperforms the other SSBs in specificity, not displaying non-specific amplification. We included a reference to recent review of approaches for enhancing isothermal amplification (Özay, B. and S.E. McCalla, A review of reaction enhancement strategies for isothermal nucleic acid amplification reactions. Sensors and Actuators Reports, 2021. 3: p. 100033). Please note this review confirms that SSBs so far have only been used to decrease non-specific binding, emphasizing the novelty of the use of gp32 in the primer annealing. 

A possible explanation is that the reagents from LSAT may have been stabilised with trehalose -without our knowledge - which may have helped the one-pot reaction, as trehalose has been demonstrated to enhance isothermal amplification reactions (Mok, E., Wee, E., Wang, Y. et al. Comprehensive evaluation of molecular enhancers of the isothermal exponential amplification reaction. Sci Rep 6, 37837 (2016). We have noted that the precise formulation is no longer provided by LSAT. Seven years ago was listed as 40 mM TrisHCl, pH 8.5 @ 25ºC, 12 mM MgCl2, 70 mM KCl, 5 mM DTT, 15% Dimethyl Sulfoxide ---- 2 mM rA,C,UTP, 1.5 mM rGTP, 0.5 mM Inosine 5'-triphosphate, 1 mM dNTP. LSAT has not mentioned a change in formulation, but simply removed the composition from the public domain. It is normal in the industry to keep the exact formulation of the buffer as a trade secret.

The performance enhancement by trehalose described in the Sci Rep article were obtained at concentrations of 0.4 M, a value that is unrealistic to be present as “contamination”, discrediting the hypothesis trehalose entered the buffer with the stabilised enzymes. 

Importantly, we have experimentally tested most of the additives listed in the newly cited review [Özay, B. and S.E. McCalla)] to facilitate the single pot reaction, identifying gp32 as the most promising candidate for NASBA. In that study, the amplification performance in the LSAT NASBA buffer was similar to that in the buffer composed using the ingredients listed seven years ago, hence we have no experimental evidence the LSAT formulation may have changed (nor do we have evidence it did not). The study did confirm an enhancement in amplification for the single pot reaction in presence of trehalose, however, the gp32 results reported here were superior. 

Considering we used the LSAT buffer as indicated in the method section of manuscript, the performance enhancement was found for gp32 but not for other SSBs, irrespective if we used the LST buffer or made our own, we continue to believe it is important to communicate our findings on gp32 to allow other researchers to elucidate and benefit from its action(s) in NASBA and other isothermal amplification methods. 

Action taken:

Added to line 104/104 

“Attenuated amplification was observed for all one step NASBA reactions.” 

We also included

“ To the authors best knowledge, there are no literature reports of single step NASBA protocols, but anecdotal reports confirmed in our laboratory suggest the reliance on the thermal annealing step varies for primer/targets.” 

Above this discussion to illustrate that the observed one-pot amplification is not unique, and has been observed, but not reported by others. 

And After line 76, the following text was added, including the abovementioned reference

“A recent review on enhancement strategies for isothermal amplifications concluded that effective additives used to eliminate the thermal denaturation step include helicase, recombinases, endonucleases, ionic liquids, betaine, proline and trehalose [15], and SSBs used for decreasing non-specific interactions and to prevent nucleic acid degradation.” 

Additionally, we replaced “eliminate” with “decrease the reliance on” to moderate the concluding statement. 

And added 

“ While mechanistic studies explaining the role of gp32 in one step NASBA, as well as to examine the effect of the target and primer are required, ..” to the conclusion 

The authors own data in the rebuttal shows another amplicon where the 1-step gives very poor signal and the gp32 addition does not rescue it to the level of the 2-step, let alone faster as they (unconvincingly from the limited data points) claim in the manuscript. I feel this proves my point, not theirs, and the gp32/1-step effect may indeed by an artifact of the assay used in the main body of the text. 2 assays are shown here, and the authors' claims are only true for 1 of them. Either more primer/sets assays should be shown where the gp32/1-step is indeed as good as the 2-step, or the authors should modify the claims in the manuscript to be more honest and in line with the data. 

Response:

We would like to thank the reviewer for pointing this out, and we understand his/her position. We however, confirmed out findings with H1N1 and the HIV gag gene, and then upon request found a strongly reduced efficacy of T4GP32 in the low copy number full HIV sample, as indicated in the first revision. We agree with Reviewer 1 that, considering we obtained this undesirable result, we should alter the manuscript to discuss this result.

Action taken:

After line 147, the following text was added:

“The effect of gp32 was studied using the Acrometrix HIV-1 control, and a near-negligible A weak improvement in ttp (from 29 min to 28.5 min) was found using this full length HIV-1 RNA primer set (Supplementary Information), a result that may be due to the low target concentration (100 copies/reaction). This more modest effect could be caused by the low sample concentration or by the decreased dependence of the HIV-1 gag gene primer set on the thermal annealing step. Improvements in ttp using gp32 were confirmed for Avian influenza A H5N1 RNA, decreasing ttp from 38 to 31 min in presence of gp32, compared with a ttp of 28 min for the two step process (unpublished results). “

That in circumstances where avoiding heat denaturation is paramount to NASBA utilization, then gp32 may help, but there may be sacrifices in performance depending on the assay. That is obviously not a barrier to high-throughput NASBA/TMA as the authors now even state, the Hologic and Biomerieux platforms do seem to work. Perhaps a simple point-of-care or at-home test NASBA would indeed be more practical if the 2-step protocol can be avoided, and the manuscript would be stronger if the authors present that more honestly.

Response: 

It is unclear to the authors what changes the reviewer would like us to make, as the 

Introduction 

“ the 2-step process could be considered inconvenient for high throughput or point of care operation as each additional processing step may present an operational and engineering challenge [14]. “ 

Discussion 

“A truly isothermal, single-step NASBA method is anticipated to enhance NASBA’s potential for adoption in portable diagnostic devices, as it eliminates the need for complex temperature and fluidic control, hence simplifying instrument and operational design. “ 

and Conclusion

“, removing an operational bottleneck by facilitating one-pot NASBA” 

all clearly state the two step process is a bottleneck for decentralised testing, and the aim of this work it so overcome this bottleneck using a SSB. 

The pandemic, and particularly Omicron wave has demonstrated that centralised PCR testing is too slow and cumbersome, and rapid antigen testing is adopted despite lower specificity, and higher false positive/false negative rates. Simplicity is the key to effective decentralised testing (see WHO ASSURED criteria), and a single step reaction is simpler than a two-step process. It does not mean the same or superior results cannot be obtained with sophisticated engineering, but if we compare it to the transport sector we have different expectations and are prepared to make different sacrifices when using a plane (complex engineering but centralised) or bicycle (less complex engineering but decentralised) for transport. 

Action taken:

To further emphasize the focus of this work towards the field of decentralised, point of care testing, we added 

“for enhanced field deployability. “ to line 91

We trust that this explanation combined with the edited text adresses the reviewers concern. 

Reviewer #3: Authors of the manuscript entitled “An Improved Nucleic Acid Sequence-Based Amplification Method Mediated by T4 Gene 32 Protein” have thoroughly revised the said manuscript in the light of comments/suggestions raised by the reviewers. Responses to reviewers’ comments have satisfactorily been addressed by the authors and have also been incorporated at the appropriate places within the revised manuscript. The revised manuscript may now be accepted for publication in PLOS ONE. However, there is a couple of very minor corrections that should be done at the Journal level before publication.

Response:

We would like to thank this reviewer for these comments

1. Introduction section on page no. 4, line no. 73- abbreviation ‘RPA’ has been used for the first time in the manuscript, so its full form ‘Recombinase Polymerase Amplification’ should also be written at that place.

Response:

Thanks for pointing out this omission, 

Action taken: 

We now included the definition before using the abbreviation.

“Compared to other isothermal amplification methods, such as LAMP or Recombinase Polymerase Amplification (RPA), the 2-step process could be considered inconvenient for high throughput or point of care operation as each additional processing step may present an operational and engineering challenge [14].

2. Results and Discussion section, page 5, line 97- ‘times to not include’ should be

‘times do not include’.

Response:

Thanks for pointing out this omission

Action taken:

The typographical error has been corrected and now reads “Please note the recorded amplification times do not include the extra time required for the addition of the amplification enzymes after thermal annealing using the 2-step protocol.”

---

## [Decision Letter · Decision Letter 2]

22 Feb 2022

PONE-D-21-08924R2An Improved Nucleic Acid Sequence-Based Amplification Method Mediated by T4 Gene 32 ProteinPLOS ONE

Dear Dr. Guijt,

Thank you for submitting your manuscript to PLOS ONE. After careful consideration, we feel that it has merit but does not fully meet PLOS ONE’s publication criteria as it currently stands. Therefore, we invite you to submit a revised version of the manuscript that addresses the points raised during the review process.

We look forward to receiving your revised manuscript.

Kind regards,

Alberto Amato

Academic Editor

PLOS ONE

Journal Requirements:

Additional Editor Comments (if provided):

The reviewer has accepted the changes made and asks for a very last modification before acceptance.

Please make the suggested change as quick as possible in order to have your manuscript published rapidly.

Reviewers' comments:

Reviewer's Responses to Questions

**Comments to the Author**

1. If the authors have adequately addressed your comments raised in a previous round of review and you feel that this manuscript is now acceptable for publication, you may indicate that here to bypass the “Comments to the Author” section, enter your conflict of interest statement in the “Confidential to Editor” section, and submit your "Accept" recommendation.

Reviewer #1: (No Response)

2. Is the manuscript technically sound, and do the data support the conclusions?

Reviewer #1: Partly

3. Has the statistical analysis been performed appropriately and rigorously? 

Reviewer #1: Yes

4. Have the authors made all data underlying the findings in their manuscript fully available?

Reviewer #1: Yes

5. Is the manuscript presented in an intelligible fashion and written in standard English?

Reviewer #1: Yes

6. Review Comments to the Author

Reviewer #1: The authors have again made improvements to the manuscript and adjusted language to be in line with what is actually demonstrated by the data. So I'm mostly in agreement it can be accepted for publication, but there are still a few issues. I said in the last review that claiming the two-step nature of NASBA is the barrier to its widespread use is unfounded, that benefit would be only in an at-home or simple device...and the authors responded with

"It is unclear to the authors what changes the reviewer would like us to make". Well here's one. The very first sentence of the manuscript in the abstract is: "The uptake of Nucleic Acid Sequence-Based Amplification (NASBA) is hindered by the requirement of a thermal denaturation step to initiate the cyclic isothermal amplification." That is plainly not true. If the authors want to cite ASSURED as justification for use of a 1-step vs. 2-step test, they should consider that the E stands for "equipment-free" which doesn't really apply to a test that uses fluorescent beacons no matter how many temperatures are involved.

If the authors would simply not claim they've fixed NASBA, a method that's used worldwide every day for diagnostics, but rather keep the claims to that they've shown maybe the denaturation step could be omitted if gp32 is added then I'd be totally fine with this manuscript. The discussion and conclusion paragraph have this much better, so just change the Abstract and I say it's okay.

7. PLOS authors have the option to publish the peer review history of their article (what does this mean?). If published, this will include your full peer review and any attached files.

Reviewer #1: No

---

## [Author Response · Author response to Decision Letter 2]

24 Feb 2022

We would like to thanks Reviewer 1 for clarifying her/his comments. 

Reviewer #1: The authors have again made improvements to the manuscript and adjusted language to be in line with what is actually demonstrated by the data. So I'm mostly in agreement it can be accepted for publication, but there are still a few issues. I said in the last review that claiming the two-step nature of NASBA is the barrier to its widespread use is unfounded, that benefit would be only in an at-home or simple device...and the authors responded with

"It is unclear to the authors what changes the reviewer would like us to make". Well here's one. The very first sentence of the manuscript in the abstract is: "The uptake of Nucleic Acid Sequence-Based Amplification (NASBA) is hindered by the requirement of a thermal denaturation step to initiate the cyclic isothermal amplification." That is plainly not true. If the authors want to cite ASSURED as justification for use of a 1-step vs. 2-step test, they should consider that the E stands for "equipment-free" which doesn't really apply to a test that uses fluorescent beacons no matter how many temperatures are involved.

Response

We would like to thank the reviewer for her/his clarification. and now understand we failed to distinguish between the ultimate goal articulated in the WHO’s Affordable, Sensitive, Specific, User-friendly, Rapid and robust, Equipment-free and Deliverable to end-users (ASSURED) criteria, and the commonly accepted reality where of point of care diagnostic products often fail to meet part of these criteria, and are not equipment free. We specifically selected reference 1 because within its focus on nucleic acid testing, it articulates the need for 

“Procedural simplicity 

In order to reduce cost, prevent malfunction and achieve robust, reliable operation within a simple package an amplification technology will preferably be a “single tube” reaction with a minimal volume, employing few reagents and few fluidic manipulations.“ (Craw, LoC, 2012)

The use of gp32 is aimed to simplify the workflow, and we acknowledge this simplification does not eliminate the need for equipment. As we now understand the standpoint of the reviewer, we amended the manuscript to prevent confusion and remove any ambiguity to the field the potential benefits of this work apply to. 

Action taken

1. Limited the field the significance of the work applies to point of care diagnostics, and improved the articulation of the scope of the work to the simplification of the work flow by modifying the opening and closing sentence of the abstract 

The uptake of Nucleic Acid Sequence-Based Amplification (NASBA) for point of care testing may be hindered by a complexity in the workflow due the requirement of a thermal denaturation step to initiate the cyclic isothermal amplification before the addition of the amplification enzymes.

….

For synthetic HIV-1 RNA, gp32 was found to improve the time-to-positivity (ttp) by average of 13.6% of one-step NASBA and 6.7% of conventional NASBA for the detection of HIV-1 RNA, showing its potential for simplifying the workflow as desirable for point of care applications of NASBA. 

2. Clarifying the purpose in line 78 by adding an additional reference to [1]

Procedural simplicity is critical reduce cost, prevent malfunction and achieve robust, reliable operation [1].

3. And re-iterating this in the conclusion

In conclusion, SSBs can be used to decrease the reliance on a thermal annealing step in NASBA, enhancing the procedural simplicity by facilitating one-pot NASBA.

If the authors would simply not claim they've fixed NASBA, a method that's used worldwide every day for diagnostics, but rather keep the claims to that they've shown maybe the denaturation step could be omitted if gp32 is added then I'd be totally fine with this manuscript. The discussion and conclusion paragraph have this much better, so just change the Abstract and I say it's okay.

Response

Now we understand the reviewers misunderstanding of our aim of simplifying the workflow vs meeting the ASSURED criteria including being equipment free, we amended the manuscript as indicated above to clarify the significance of this work is limited to the workflow, and will not allow NASBA to meet he ASSURED criteria. In response to the reviewers wish to amend the abstract, we provide the amended abstract below. 

Action taken: 

Amended abstract (new text in red) 

The uptake of Nucleic Acid Sequence-Based Amplification (NASBA) for point of care testing may be hindered by a complexity in the workflow due the requirement of a thermal denaturation step to initiate the cyclic isothermal amplification before the addition of the amplification enzymes. Despite reports of successful enhancement of other DNA and RNA amplification methods using DNA and RNA binding proteins, this has not been reported for NASBA. Here, three single-stranded binding proteins, RecA, Extreme Thermostable Single-stranded binding protein (ET SSB) and T4 gene gp32 protein (gp32), were incorporated in NASBA protocol and used for single pot, one-step NASBA at 41 °C. Indeed, all SSBs showed significantly improved amplifications compared with the 2-step process, but only gp32 showed no non-specific aberrant amplification, and slightly improved the time-to-positivity in comparison with the conventional NASBA. For synthetic HIV-1 RNA, gp32 was found to improve the time-to-positivity (ttp) by average of 13.6% of one-step NASBA and 6.7% of conventional NASBA for the detection of HIV-1 RNA, showing its potential for simplifying the workflow as desirable for point of care applications of NASBA.

---

## [Editor Report · Decision Letter 3]

2 Mar 2022

An Improved Nucleic Acid Sequence-Based Amplification Method Mediated by T4 Gene 32 Protein

PONE-D-21-08924R3

Dear Dr. Guijt,

We’re pleased to inform you that your manuscript has been judged scientifically suitable for publication and will be formally accepted for publication once it meets all outstanding technical requirements.

Kind regards,

Alberto Amato

Academic Editor

PLOS ONE

Additional Editor Comments (optional):

After the third round of revision I am pleased that the Authors and the Reviewer have found a point of agreement that allows the acceptance of the manuscript to proceed.
---

## [Editor Report · Acceptance letter]

16 Mar 2022

PONE-D-21-08924R3 

An Improved Nucleic Acid Sequence-Based Amplification Method Mediated By T4 Gene 32 Protein 

Dear Dr. Guijt:

I'm pleased to inform you that your manuscript has been deemed suitable for publication in PLOS ONE. Congratulations! Your manuscript is now with our production department. 

Kind regards, 

on behalf of

Dr. Alberto Amato 

Academic Editor

PLOS ONE